# Vaginal Toxicity Management in Patients with Locally Advanced Cervical Cancer following Exclusive Chemoradiation—A Nationwide Survey on Knowledge and Attitudes by the Italian Association of Radiotherapy and Clinical Oncology (AIRO) Gynecology Study Group

**DOI:** 10.3390/medicina59020385

**Published:** 2023-02-16

**Authors:** Francesca De Felice, Lisa Vicenzi, Gabriella Macchia, Andrea Vavassori, Elisabetta Perrucci, Annamaria Cerrotta, Valentina Lancellotta, Sofia Meregalli, Lorena Draghini, Antonietta Augurio, Vitaliana De Sanctis

**Affiliations:** 1Radiation Oncology, Policlinico Umberto I, Department of Radiological, Oncological and Pathological Sciences, “Sapienza” University of Rome, Viale Regina Elena 326, 00161 Rome, Italy; 2Operative Research Unit of Radiation Oncology, Fondazione Policlinico Universitario Campus Bio-Medico, Via Álvaro del Portillo, 200, 00128 Rome, Italy; 3Radiation Oncology Unit, Gemelli Molise Hospital, Università Cattolica del Sacro Cuore, Largo Agostino Gemelli, 1, 86100 Campobasso, Italy; 4Department of Radiotherapy, European Institute of Oncology IRCCS, Via Giuseppe Ripamonti, 435, 20141 Milano, Italy; 5Radiation Oncology Section, Perugia General Hospital, Piazzale Giorgio Menghini, 3, 06129 Perugia, Italy; 6Radioterapia Oncologica, Fondazione IRCS, Istituto Nazionale dei Tumori di Milano, Via Giacomo Venezian, 1, 20133 Milano, Italy; 7U.O.C. Radioterapia Oncologica, Dipartimento di Diagnostica per Immagini, Radioterapia Oncologica ed Ematologia, Fondazione Policlinico Universitario A. Gemelli IRCCS, Via Giuseppe Moscati, 31, 00168 Rome, Italy; 8Radioterapia Ospedale San Gerardo-ASST Via G. B. Pergolesi, 33, 20900 Monza, Italy; 9Radiotherapy Oncology Centre, S. Maria Hospital, Viale Tristano di Joannuccio, 05100 Terni, Italy; 10Radiation Oncology Unit, “SS Annunziata” Hospital, “G. D’Annunzio” University, Via dei Vestini, 66100 Chieti, Italy; 11Department of Medicine, Surgery and Translational Medicine, “Sapienza” University of Rome, Radiotherapy Oncology, St. Andrea Hospital, Via di Grottarossa 1035, 00189 Rome, Italy

**Keywords:** vaginal toxicity, locally advanced cervical cancer, survey

## Abstract

*Background and Objective*: Exclusive radiotherapy, including external beam radiotherapy (EBRT) and interventional radiotherapy/brachytherapy (IRT/BT), with concurrent cisplatin-based chemotherapy, represents the standard of care in patients with locally advanced cervical cancer (LACC). The emerging topic of vaginal toxicity has become a key endpoint in LACC management, although different approaches and non-standardized procedures were available. Our aim was to analyze a nationwide study of the attitudes of Italian gynecological radiation oncology teams in the management of LACC patients’ vaginal toxicities. *Methods*: A nationwide survey of radiation oncologists specializing in the treatment of gynecological malignancies was performed, using the free SurveyMonkey platform, consisting of 26 items. The questionnaire was proposed by the Italian Association of Radiation Oncologists (AIRO) gynecological working group to all 183 Italian radiation oncology institutions, as per AIRO’s website. Results: Fifty-eight questionnaires (31%) were completed and returned. The assessment of acute and late vaginal toxicities was systematic in 32 (55.2%) and 26 (44.8%) centers, respectively. In the case of EBRT, 70.7% of centers, according to the contouring and treatment plan data, did not contour the vagina as an organ at risk (OAR). Vaginal dose constraints were heterogeneous for both EBRT and IRT/BT. Local treatment to prevent vaginal toxicity was prescribed by 60.3% of radiation oncologists, mostly vaginal hyaluronic acid cream, and one center recommended vaginal estrogen preparations. During follow-up visits, vaginal toxicity was considered an issue to be investigated always (*n* = 31) or in sexually active women only (*n* = 11). *Conclusions*: This survey showed that wide variation exists with regard to recording and treating vaginal toxicity after exclusive chemoradiation for cervical cancer, underscoring the need to develop more comprehensive guidelines for contouring e-dose reporting of the vagina, so as to implement clinical approaches for vaginal toxicity.

## 1. Introduction

Exclusive radiotherapy, including external beam radiotherapy (EBRT) (45–50 Gy, 1.8 Gy per fraction) and interventional radiotherapy/brachytherapy (IRT/BT) (40–45 Gy equieffective dose to 2 Gy (EQD2) per fraction to reach a total EBRT + IRT/BT dose of ≥85–90 Gy EQD2), with concurrent weekly cisplatin-based chemotherapy, is the treatment of choice for most patients with locally advanced cervical cancer (LACC) [1,2,3]. Despite, over the years, clear guidelines that have been developed to address specifics of organs at risk (OARs) and target volume definitions, independently of radiotherapy techniques [4,5,6,7,8,9,10,11,12], little attention has been paid to the resultant vaginal toxicity that women could experience during follow-up in the short and long term. This aspect was partly left out due to the historical idea that the vagina was considered a radioresistant organ; however, a differential radiosensibility was recognized for the upper vagina compared to the middle-inferior vagina, the latter being more radiosensible [13,14,15,16].

Recently, attention has been focused on dose reporting for the vagina, as recommended by ICRU 89 and by international prospective observational studies on MRI-guided brachytherapy in locally advanced cervical cancer (EMBRACE II) protocol, which suggested a dose limit in the upper vagina (ICRU rectovaginal point as reference) lower or equal to a 65 Gy equivalent dose in a 2 Gy fraction (EQD2) (α/β = 3 Gy) [17,18]. Nevertheless, until now, there have been relevant discrepancies about vaginal contouring and vaginal dose reporting in clinical practice, and still no consensus was has been reached about a standardized correlation between dose to vaginal mucosa and vaginal toxicities onset [19].

For the vagina, these toxicities historically included necrosis, vescico/rectovaginal fistula, while, in the past decade, vaginal stenosis and, above all, dyspareunia were the most frequently experienced symptoms in LACC patients who had undergone exclusive (chemo)radiotherapy and IRT. In fact, vaginal stenosis was recorded for from 21% to 38% of LACC patients, while dyspareunia ranged from 50–60% to 80% of the patients [18,20,21,22].

Radiation-induced vaginal stenosis and dyspareunia may have a substantial adverse impact on patient wellbeing due to the impairment of sexual quality of life, and it represents a dysfunction that is likely underreported [23]. Although sexuality, recognized as a human right and being a key component of quality of life, represents an important public-health issue in the setting of long-term-survivor LACC patients, the integration of sexuality management in routine care is frequently lacking [24,25,26].

Prevention strategies may include vaginal dilatation, topical therapies, and laser therapy, but consensus is not reached due to the paucity of high-level evidence regarding the prevention and management of vaginal stenosis/dyspareunia [20,21,27].

Therefore, vaginal toxicities represent a gray area in clinical practice as well as in research and educational programs.

In particular, data about patterns of care surveys on radiation-induced vaginal toxicity are scarce and, to the best of our knowledge, in Italy, no information is available on a national radiation oncologist approach to vaginal toxicity after chemoradiation in LACC patients.

For the first time, a nationwide study was carried out with the aim of describing the practices of Italian gynecological radiation oncology teams in the management of LACC patients’ vaginal toxicities in terms of knowledge, recording, and prevention. The subsequent aim could be to become proactive in suggesting collaborative multicenter trials.

## 2. Materials and Methods

### 2.1. Survey Design and Questionnaire

The survey was developed by two authors (VDS and FDF) in October 2021, reviewed by the AIRO gynecological (GM, EP, AC, VL, SM, LD, and AA) and interventional radiotherapy (AV and LV) working group representatives, and approved by the AIRO Scientific Committee.

The main goal of the questionnaire was to enquire into local knowledge, treatment standards, and attitudes for the management of vaginal toxicity after exclusive/definitive chemoradiation in LACC patients. The questionnaire, consisting of 26 items, was grouped into three sessions: (i) data of the participating physicians (4 questions): gender, geographic location, professional site (e.g., university or community hospitals or private practices), and work methods (presence or absence of a multidisciplinary group); (ii) clinical information (18 questions): number of LACC patients treated per year, rate of acute and late toxicity, radiotherapy planning issues, specific supportive care, and attitude toward considering vaginal toxicity as a clinical issue; (iii) research information (4 questions): availability to join retrospective/prospective studies on the issue and the need for guidelines/consensus conference/contouring courses. We only analyze single institution vaginal toxicity management during/after standard treatment of care (EBRT + IRT/BT). Details on treatment options, such as differences in patient populations, treatment protocols, and survival outcomes, were not combined. The vast majority of questions were radio-button-type questions, and respondents were asked to choose a single answer from a list of available options. A minority were open-ended questions. The questionnaire was designed to be completed in approximately 10 min. 

### 2.2. Participating Physicians

A nationwide survey of radiation oncologists specializing in the treatment of gynecological malignancies was carried out. The participating centers were not pre-selected. The questionnaire was proposed by the Italian Association of Radiation Oncologist (AIRO) gynecological working group to all 183 Italian radiation oncology institutions as per AIRO’s website (www.radioterapiaitalia.it: accessed on 16 April 2022). For each center exclusively, the radiation oncologist in charge of the gynecological team was required to fill out the form with the aim of capturing the specialist attitude and pattern of care. The survey was strictly confidential and was available online for 50 days, from 11 May to 30 June 2022. The questionnaire was accessible from a computer, tablet, or smartphone using the free SurveyMonkey platform (www.SurveyMonkey.com accessed on 7 July 2022). Two reminder emails were sent to increase the number of responses.

### 2.3. Statistical Analysis

Returned questionnaires were collected centrally at Sapienza University of Rome and data were entered into an electronic database. The data processing occurred in July 2022. Standard descriptive statistics were used to evaluate the distribution of responses. Data were reported as means or percentages. Qualitative data were reported by identifying key themes and reporting direct quotes. Statistical analysis was carried out using RStudio 0.98.1091 software.

## 3. Results

Fifty-eight questionnaires, accounting for 31.7% of the 183 Italian centers available on the AIRO website, were completed and returned.

### 3.1. Demographic Information

Most of the participating physicians came from community hospitals (*n* = 33), followed by university (*n* = 14) and private (*n* = 11) hospitals. The responding centers were evenly distributed throughout the country, with a predominance in the northern region (34 responses, 58.7%) (Figure 1).

The radiation oncologists in charge of gynecological cancer management were mostly female (*n* = 41, 70.7%) and 49 (84.5%) respondents referred their LACC patients to their own institutional multidisciplinary tumor board for shared clinical decisions.

### 3.2. Clinical Information

We next investigated the core of radiation oncologist approaches in terms of number of patients per year (2020), rate of acute and late toxicity, radiotherapy planning issues, specific supportive care, attitude to consider vaginal toxicity as a clinical issue, and patient information sharing.

Stratification of participating radiation oncology centers according to the number of treated patients per year is reported in Figure 2.

Notably, only one center reported receiving more than 100 LACC referrals per year, whereas the vast majority (76%) handle fewer than 20 patients per year.

The rate of vaginal toxicities according to respondents is shown in Figure 3.

Overall, most centers reported some percentage of patients who experienced acute and/or late vaginal toxicity, with 2% of experts unable to quantify the data.

In particular, the assessment of acute and late vaginal toxicities was systematic in 32 (55.2%) and 26 (44.8%) centers, respectively. For acute toxicity, the Common Terminology Criteria for Adverse Events (CTCAE) and Radiation Therapy Oncology Group/European Organization for Research and Treatment of Cancer (RTOG/EORTC) scales were used by 14 (43.8%) and 17 (53.1%) respondents, respectively. For late toxicity, RTOG/EORTC and Late Effects Normal Tissues/Subjective Objective Management Analytic (LENT/SOMA) scales were used by 20 (76.9%) and 5 (19.2%) respondents, respectively. Of note, one center did not use any toxicity scale to grade both acute and late vaginal toxicity.

In the case of EBRT, the majority of centers (*n* = 41, 70.7%), according to the contouring and treatment plan data, did not contour the vagina as OAR. The remainder (*n* = 17, 29.3%) either always (*n* = 8) or occasionally (*n* = 9) delineated the vagina. Of these 17 institutions, radiation oncologists prioritized conformity overdose to the vagina either always (*n* = 4) or occasionally (*n* = 8). Moreover, in five cases, the vagina was contoured as OAR but not considered during plan approval.

Vaginal dose constraints were heterogeneous as follows: mean dose (Dmean) less than 40 Gy (two respondents), Dmean less than 15 Gy (one center), Dmean less than 43 Gy only for exclusive treatments with simultaneous integrated boost (one center), Dmax less than 70 Gy (two respondents), less than 50% of the vaginal volume that received 20 Gy (V20 < 50%), V30 < 30%, V40 < 5% (one center), and five respondents did not specify dose constraints.

In the case of IRT/BT, nine respondents (15.5%)—either always (*n* = 7) or occasionally (*n* = 2)—delineated the vagina as OAR, and they always (*n* = 4) or sometimes (*n* = 4) considered dose to vagina during plan approval. In one case, vagina was defined as OAR but it was not prioritized during plan approval. Dose constraints reported that vaginal surface ranged from 90 Gy to 175 Gy among respondents.

Thus, 40% and 60% of respondents reported acute and late vaginal toxicity, respectively, ranging from 0% to 40% in the investigated setting.

In 33 sites (56.9%), according to the attitudes of the specialists, patients were routinely warned about the possibility of acute/late vaginal toxicities, while nine responders (15.5%) said they at least occasionally did so. Most radiation oncologists (*n* = 35, 60.3%) prescribed local treatment to prevent vaginal toxicity: 34 teams recommended vaginal hyaluronic acid cream and 1 center recommended vaginal estrogen preparations. During follow-up visits, vaginal toxicity was considered an issue to be investigated always (*n* = 31) only in sexually active women (*n* = 11).

Centers were slightly less likely to support vaginal care information sharing for patients (*n* = 27, 46.6%) than update. The trend was similar concerning the updating of vaginal care guidelines/protocols for staff (*n* = 23, 39.7%)

### 3.3. Research Information

Respondents were asked how they thought education and training initiatives could best encourage support for sharing information. Clinical guidelines (68.2%), contouring course (18.2%), and expert consensus (13.6%) were the preferred methods to increase clinical management of these toxicities. Most teams declared interest in participating in a retrospective (*n* = 30, 51.7%) or prospective (*n* = 37, 63.8%) clinical study on this topic.

## 4. Discussion

Long-term vaginal changes, such as decreased lubrication and vaginal stenosis, can occur in LACC patients who have undergone exclusive chemoradiation, with acute and late toxicity due to damage to the vaginal epithelium, connective tissue, and small blood vessels as a first step. As a second step, tissue hypoxia due to inflammation with local cell death and reduced local blood flow takes over. As a final step, the loss of elastin, collagen deposition, and hyalinization impair the vaginal mucosa, causing a loss of lubrication and, ultimately, fibrosis [23,28].

Particularly, vaginal toxicities can cause difficulties during sexual intercourse, which can have detrimental physiological and psychological implications on the patient’s quality of life [29,30].

Dyspareunia occurs in up to 80% of patients with a relevant negative impact on sexuality, mainly due to pain during vaginal intercourse, while sexual satisfaction seems not to be significantly impaired [20,21,26]. Despite the relevant impairment in quality of life, only 10–28% of cervical cancer patients received information about the onset of this kind of acute and, above all, late vaginal toxicity [26]. In fact, during the follow-up visits of women treated with pelvic radiotherapy, late symptoms were assessed for bowel (81%) and bladder (70%), while vaginal toxicity was explored less frequently (42%); sexual issues were broached in only 25% of patients [31]. Because of changes in health-related quality of life in LACC survivors, vaginal toxicities represent a current unmet need that have to be integrated into our clinical practice during all the steps of diagnosis, treatment, and follow-up.

Interventions for vaginal toxicities prevention are based on limited scientific evidence and, even in the international context, there is no standard strategy guiding prevention of this condition. Only recently, some practical recommendations are beginning to focus on the vaginal/sexual morbidity aspect [20,21,27,32,33,34,35,36]. First of all, attention has been focused to consider the vagina as an organ at risk, according to ICRU89 and EMBRACE I indications, although a consensus has still not been reached for contouring and for reporting the dose to the vagina [17,18]. Moreover, no consensus has been reached about interventions for the prevention of vaginal toxicities, although some indications can be drawn regarding the use of topical hyaluronic acid, laser therapy, dilators, and hormone replacement therapy [20,32,35,36].

The aim of this survey was to explore the attitude of Italian Radiotherapist Oncologists for the recording and prevention of vaginal toxicities in patients with LACC treated with exclusive pelvic radiotherapy and IRT/BT as a boost.

The responding centers came mainly from community hospitals, followed by university and private hospitals, with a predominance in the northern region, reflecting the distribution of Italian radiotherapy institutions, as expected. We also observed a predominance of female radiation oncologists involved in multidisciplinary tumor boards.

From our analysis, we recorded a large number of centers that treated fewer than 20 patients/year. This clinical scenario may be due to the therapeutic approaches of LACC patients in Italy that include a neoadjuvant chemotherapy and/or surgery strategy, so the numbers of LACC patients that underwent exclusive radiochemotherapy may be lower than expected.

Systematic assessment and recording of acute and late vaginal toxicity are of paramount importance but were routinely performed and reported by roughly half a percent of the Italian centers. With a range from 0% to 40% of toxicities reported, this issue has to be considered challenging in LACC patients, as also reported in other series [21,32]. Overall, the rate of vaginal stenosis was 60% in LACC patients, as reported by Varitè et al., depending on the length of treated vaginal canal, total dose, and dose per fraction. Kirchheiner et al. reported that vaginal functioning problems (dryness, shortening, tightening, and dyspareunia) ranged from 9% to 22% in LACC patients treated with MRI-based IRT, also with compromised enjoyment in 37% at baseline and in 47% during the follow-up [21,32].

Common scales, such as CTCAE and RTOG/EORTC, were mostly used for recording acute toxicities, whereas the RTOG/EORTC scale was preferred for late toxicities. Notably, the modern Patient Reported Outcome (PRO) scale was not used, although PRO scales may be more effective with particular regard for impairments in sexual activity [32].

The vagina was not routinely considered as OAR, especially during the EBRT contouring and treatment planning phase, probably due to the lack of firm constraints in clinical practice. In particular, a very low rate of vaginal contouring during the EBRT phase was recorded by our survey and, when contoured frequently, dose to vagina was not considered during the planning phase. As shown in the literature, no standard dose constraint for vaginal toxicity was available. Historical studies on the use of low dose rate BRT report a dose of 140–150 Gy as a tolerance dose [13]. Its lower portion appears to be more sensitive to radiation, showing tolerance levels not exceeding 98 Gy [15]. Recently, the authors of the EMBRACE studies, in consideration of the absence of constraint evidence based on the dose to be administered to the vagina, proposed a maximum dose constraint at the rectovaginal point < 65 Gy in EQD2, a dose at which they observed a rate of severe vaginal stenosis (G > 2) less than 20% [18,27].

Indeed, in our survey, large variation in vaginal dose constraints was adopted and reported and this scenario was similar to when IRT/BT was used as a boost. We believe that the historical and consolidated attitude of not considering the vagina as potentially an OAR, the few constraints and toxicity data from less recent studies, and the too-recent data from EMBRACE are the causes of this variability and lack of attention.

Nearly 60% of centers used to inform patients regarding the occurrence of vaginal toxicity. The lacking 40% may also be partly explained by the lack of biomedical interventions developed to treat female sexual difficulties, as compared to those available for the management of erectile dysfunction [31]. However, topical hyaluronic acid was the most frequently prescribed local treatment, rarely followed by a topical estrogen preparation. No vaginal dilatators or laser therapy treatment were reported, even if there is some evidence in the literature [34,35]. Perrone et al., in their prospective observational study on laser therapy, noticed an improvement in vaginal length and Vaginal Health Index in patients submitted to radiotherapy [34]. In their review, Damast et al. carefully analyzed different aspects concerning vaginal stenosis and the common application of vaginal dilator therapy to prevent this vaginal morbidity in not reporting univocal results, mainly due to the low adherence rate of GYN cancer survivors to vaginal dilator therapy recommendations [35].

Moreover, systemic estrogen therapy was also employed in this setting of patients with the aim to reduce vaginal toxicity, but no center reported this approach [32]. It should be noted that this finding may be underestimated because the study was addressed by radiation oncologists and systemic hormone therapy is often supervised by other professionals.

In our survey, vaginal toxicity was considered to be an issue that should be investigated in all patients, and not only in sexually active patients, because vaginal toxicity may also have an impact on the occurrence of urinary symptoms or pelvic disease [21].

Finally, nearly 60% of centers were interested in participating in retrospective or prospective clinical study recording and managing vaginal toxicities; nevertheless, a quite stringent need for updated clinical guidelines for patient information and vaginal care procedures emerged.

It is important to take into account the survey’s strengths and drawbacks. The low response rates (31.7%) raise the potential for non-respondent bias, limiting generalizability and precluding straightforward comparisons (low confidence).

However, we can also consider that, with the IRT/BT being a special technique that is not available in all radiotherapy centers in Italy, 31.7% of the responders out of 183 Italian RT centers can reflect a high adherence of Italian IRT/BT centers to our survey, while the institutions that did not have IRT/BT facilities most likely did not respond. Therefore, although the response rate was not as high as expected, our cohort of respondents reflected Italian radiation oncologists with expertise in the field of vaginal toxicity. Finally, we can observe that vaginal toxicities were assessed in nearly 50% of LACC patients during the follow-up controls in our survey. Furthermore, modern scales that focus more on sexual activity, such as PRO, have been overlooked. Lastly, we have to consider that vaginal toxicities and their impact on quality of life are sometimes not reported by patients themselves, mainly due to it being a condition of psychological discomfort.

These findings underscore the need to develop more effective clinical decision support and targeted clinician education to address knowledge gaps and entrenched practices in vaginal toxicity issues.

## 5. Conclusions

The present study showed that extensive variation exists with regard to recording and treating vaginal toxicity after exclusive chemoradiation for cervical cancer, though it is a clinically relevant problem. Currently, there is a lack of international and national recommendations or guidelines; nevertheless, a favorable attitude toward cooperative studies emerged from the national survey. Further efforts in primary prevention and effective therapies are needed, passing through the identification of a more comprehensive Toxicity Scale and additional retrospective/prospective multi-center studies.

## Figures and Tables

**Figure 1 medicina-59-00385-f001:**
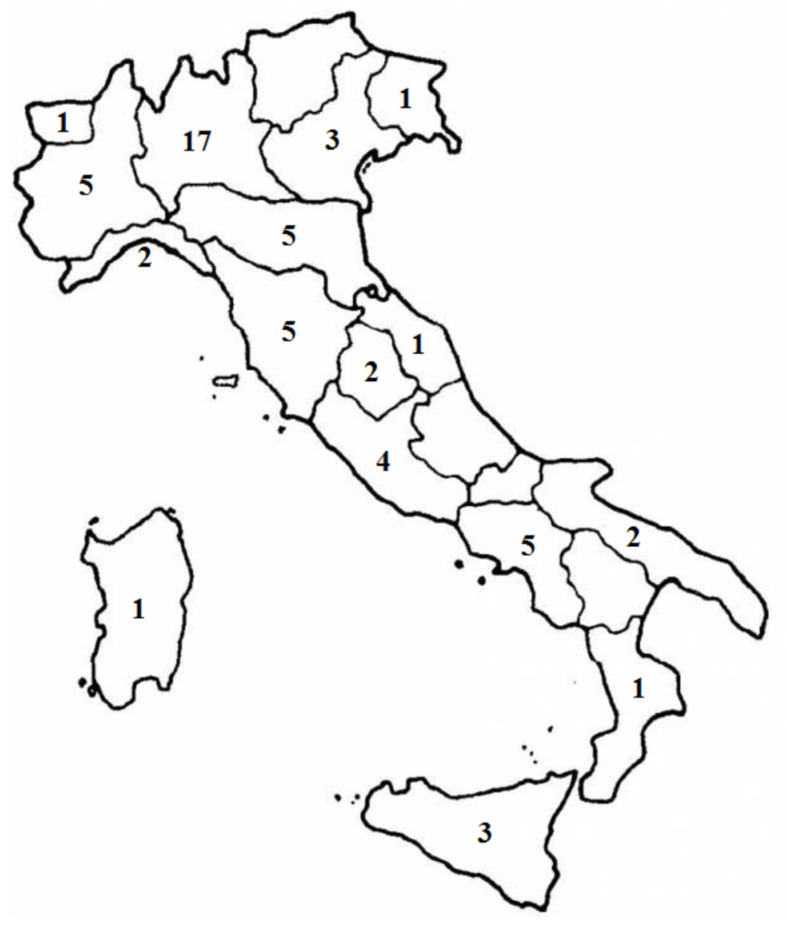
Distribution of centers (radiation oncologists who are experts in gynecological malignancy management) that completed the survey.

**Figure 2 medicina-59-00385-f002:**
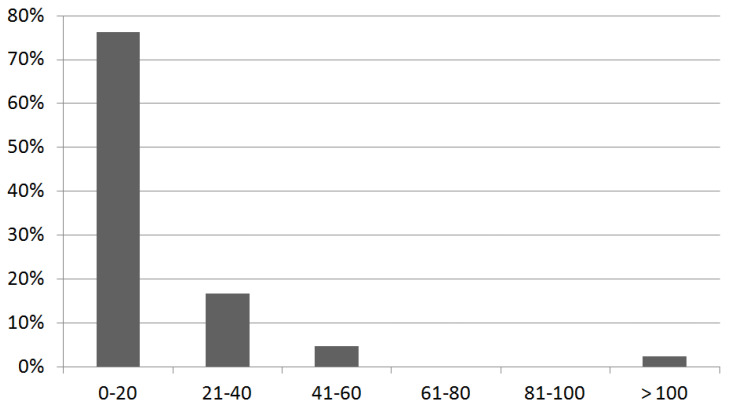
Distribution of the Italian LACC radiation oncology teams who participated in the national survey according to the number of LACC treated per year. The *x*-axis refers to the number (range) of treated LACC patients per year in the center; the *y*-axis refers to the center. For instance, the first bar indicates that approximately 80% of centers (*y*-axis) treats 0–20 LACC patients per year (*x*-axis).

**Figure 3 medicina-59-00385-f003:**
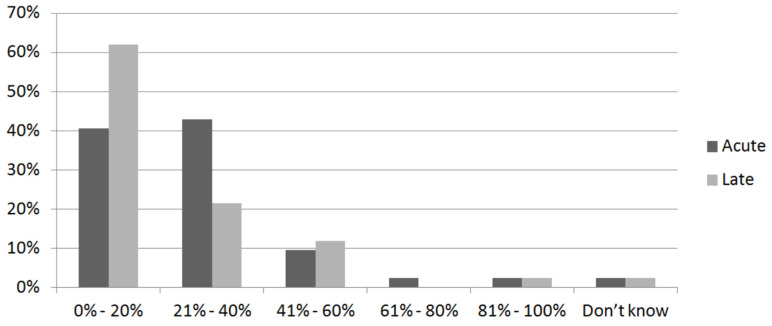
Distribution of reported acute and late vaginal toxicity. The *x*-axis refers to the percentage of patients (range) who referred vaginal toxicity; the *y*-axis refers to the center. For instance, the first two bars indicate that approximately 40% and 60% of centers (*y*-axis) report up to 20% of patients (*x*-axis) who experienced acute (dark-gray bar) and late (light-gray bar) vaginal toxicity, respectively.

## Data Availability

Not applicable.

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
