# Peer review of "Vaginal Toxicity Management in Patients with Locally Advanced Cervical Cancer following Exclusive Chemoradiation—A Nationwide Survey on Knowledge and Attitudes by the Italian Association of Radiotherapy and Clinical Oncology (AIRO) Gynecology Study Group"

_medicina, 2023, doi:10.3390/medicina59020385_

Round 1
Reviewer 1 Report
The study is aimed at studying treatment standards for vaginal toxicity post chemoradition. The authors have collected survey results from various health centers in order to study the treatment options and they summarize the results in the paper, discussing the results in detail.
The major concern with the manuscript is the novelty aspect of it. The authors share survey results where they have summarized descriptive statistics in the form of mean and percentages. However, it is commonly recognized that the treatment options may vary in the country of study.
The novelty aspect of the study could be stressed better by describing how the disease management is being done at various centers given the different patient conditions(including the various histological conditions). This will help come up with method recommendations for establishing guidelines. Also it will be interesting to see if treatment conditions are varying consistently by clinical outcomes of the chemoradiation.
In addition the authors could rank the commonly used treatment methods across various surveyed centers and suggest to readers that what are mostly widely used methods and provide their objective views on the methods accordingly.
The authors bring up the concern of low response rate and also mention two email reminders were sent. As authors characterized, roughly half the centers responded, but due to the abundance of non-responders, the conclusions have low confidence statistically.
Author Response
The study is aimed at studying treatment standards for vaginal toxicity post chemoradition. The authors have collected survey results from various health centers in order to study the treatment options and they summarize the results in the paper, discussing the results in detail. The major concern with the manuscript is the novelty aspect of it. The authors share survey results where they have summarized descriptive statistics in the form of mean and percentages. However, it is commonly recognized that the treatment options may vary in the country of study. The novelty aspect of the study could be stressed better by describing how the disease management is being done at various centers given the different patient conditions(including the various histological conditions). This will help come up with method recommendations for establishing guidelines. Also it will be interesting to see if treatment conditions are varying consistently by clinical outcomes of the chemoradiation. In addition the authors could rank the commonly used treatment methods across various surveyed centers and suggest to readers that what are mostly widely used methods and provide their objective views on the methods accordingly.
- Text has been modified.
The authors bring up the concern of low response rate and also mention two email reminders were sent. As authors characterized, roughly half the centers responded, but due to the abundance of non-responders, the conclusions have low confidence statistically.
- Text has been modified.
Reviewer 2 Report
This paper presents that wide variation exists with regard to recording and treating vaginal toxicity after exclusive chemoradiation for cervical cancer. This topic is interesting and deserves a constructive discussion.
However, I concern some minor issues as listed below for publication.
I hope that my comment is useful for the improvement of the article.
Minor concerns:
1. I found Figure 2 and 3 are difficult to interpret. I think the format of the graphs need to be changed or the legends need to be described in more detail.
Author Response
I found Figure 2 and 3 are difficult to interpret. I think the format of the graphs need to be changed or the legends need to be described in more detail.
- Legends have been modified.
Reviewer 3 Report
 This study is a questionnaire about how vaginal toxicity is managed after exclusive chemoradiation for locally advanced cervical cancer. The study seemed to be of great interest to many readers because vaginal problems after radiotherapy are often difficult to discuss openly with patients because of the sexual nature, and there has been no standardization of how to deal with them. However, there were some points that could be modified to make the paper better.
1. In the abstract and discussion section, there is a statement "deserves a mounting attention," but I thought it is better to omit this statement because it is not scientific. As noted in the discussion section, the low response rate of 31% is the main limitation of the study.
Wouldn't a low response rate be inconsistent with "deserves a mounting attention"? It is possible that the respondents to this survey were biased toward radiation oncologists who are interested in vaginal toxicity.
Therefore, it does not seem appropriate to conclude that there is an increased interest in vaginal toxicity because a higher percentage of patients in this study were routinely warned about the occurrence of vaginal toxicity than past studies.
2. Also related to (1) above, the percentage of female radiation oncologists managing gynecologic cancers is 70.7%, which is high. Are female radiation oncologists more often in charge of gynecological cancer in Italy? Or were there specially many women among the respondents of this survey? Please clarify.
3. Figure 3 is confusing because both x- and y-axis units are "%". Please add labels to make the figure clearer.
Author Response
This study is a questionnaire about how vaginal toxicity is managed after exclusive chemoradiation for locally advanced cervical cancer. The study seemed to be of great interest to many readers because vaginal problems after radiotherapy are often difficult to discuss openly with patients because of the sexual nature, and there has been no standardization of how to deal with them. However, there were some points that could be modified to make the paper better.
- In the abstract and discussion section, there is a statement "deserves a mounting attention," but I thought it is better to omit this statement because it is not scientific. As noted in the discussion section, the low response rate of 31% is the main limitation of the study. Wouldn't a low response rate be inconsistent with "deserves a mounting attention"? It is possible that the respondents to this survey were biased toward radiation oncologists who are interested in vaginal toxicity. Therefore, it does not seem appropriate to conclude that there is an increased interest in vaginal toxicity because a higher percentage of patients in this study were routinely warned about the occurrence of vaginal toxicity than past studies.
- Text has been modified.
- Also related to (1) above, the percentage of female radiation oncologists managing gynecologic cancers is 70.7%, which is high. Are female radiation oncologists more often in charge of gynecological cancer in Italy? Or were there specially many women among the respondents of this survey? Please clarify.
- It is just a descriptive result.
- Figure 3 is confusing because both x- and y-axis units are "%". Please add labels to make the figure clearer.
- Legend has been modified.
Round 2
Reviewer 1 Report
Thank you for considering recommendations carefully and including appropriate edits to state claims. The claims are acceptable in the current form and reads well.